# Genetic Engineering of *Talaromyces marneffei* to Enhance Siderophore Production and Preliminary Testing for Medical Application Potential

**DOI:** 10.3390/jof8111183

**Published:** 2022-11-09

**Authors:** Artid Amsri, Somdet Srichairatanakool, Aphiwat Teerawutgulrag, Sirida Youngchim, Monsicha Pongpom

**Affiliations:** 1Department of Microbiology, Faculty of Medicine, Chiang Mai University, Chiang Mai 50200, Thailand; 2Department of Biochemistry, Faculty of Medicine, Chiang Mai University, Chiang Mai 50200, Thailand; 3Department of Chemistry, Faculty of Science, Chiang Mai University, Chiang Mai 50200, Thailand

**Keywords:** *Talaromyces marneffei*, Δ*sreA*, siderophore, coprogen B, iron chelator

## Abstract

Siderophores are compounds with low molecular weight with a high affinity and specificity for ferric iron, which is produced by bacteria and fungi. Fungal siderophores have been characterized and their feasibility for clinical applications has been investigated. Fungi may be limited in slow growth and low siderophore production; however, they have advantages of high diversity and affinity. Hence, the purpose of this study was to generate a genetically modified strain in *Talaromyces marneffei* that enhanced siderophore production and to identify the characteristics of siderophore to guide its medical application. SreA is a transcription factor that negatively controls iron acquisition mechanisms. Therefore, we deleted the *sreA* gene to enhance the siderophore production and found that the null mutant of *sreA* (Δ*sreA*) produced a high amount of extracellular siderophores. The produced siderophore was characterized using HPLC-MS, HPLC-DAD, FTIR, and ^1^H- and ^13^C-NMR techniques and identified as a coprogen B. The compound showed a powerful iron-binding activity and could reduce labile iron pool levels in iron-loaded hepatocellular carcinoma (Huh7) cells. In addition, the coprogen B showed no toxicity to the Huh7 cells, demonstrating its potential to serve as an ideal iron chelator. Moreover, it inhibits the growth of *Candida albicans* and *Escherichia coli* in a dose-dependent manner. Thus, we have generated the siderophore-enhancing strain of *T*. *marneffei*, and the coprogen B isolated from this strain could be useful in the development of a new iron-chelating agent or other medical applications.

## 1. Introduction

Iron is an essential nutrient for all forms of life. Iron plays an important function in different cellular processes, such as DNA metabolism, protein function, fatty acid synthesis, and other chemical reactions. Iron is a transition metal that is present in two oxidation states, ferrous iron (Fe^2+^) and ferric iron (Fe^3+^). Generally, the ferric form of iron is insoluble in an aerobic environment, making it largely unavailable to organisms. Microorganisms have developed mechanisms to solubilize and take up ferric iron by producing high-affinity iron acquisition molecules. Siderophores are metal-chelating agents with low molecular weight produced by bacteria and fungi; they specifically bind the iron and deliver it into the microbial cells through specific receptors [1]. Although the basic role of siderophores is to provide soluble iron to microbes for their growth, they have also received much attention because of their potential roles in medical fields. Examples include the developments of the Trojan-horse strategy and diagnostic assays and their use as iron chelators in treatment of patients with iron overload.

Siderophores have been applied as a “Trojan horse” in therapeutic uses. Complexes between siderophore and antimicrobial can facilitate the delivery of antibiotics to pathogens. Deferoxamine (DFO)-nalidixic acid conjugate and spermidine-based bis-catechol-cephalosporin conjugate have been shown to inhibit the growth of pathogenic bacteria [2]. In addition, siderophores have been used as the diagnostic marker for the detection of some infectious diseases. Gallium (III)-DFO (Gallium-67 radiolabeling) is used as a tool for imaging the location of *Staphylococcus aureus* infection in mice [3]. In the treatment of iron overloaded diseases, DFO which is a hexadentate siderophore isolated from *Streptococcus pilosus* bacteria is a commercial FDA-approved drug [4]. Importantly, DFO monotherapy and the combination with other iron chelators has the property to decrease levels of plasma redox iron (such as non-transferrin bound iron and labile plasma iron) and ameliorate oxidative stress in thalassemia patients with iron overload [5,6,7]. However, DFO has some side effects and toxicity for dermatological, ocular, and auditory systems [8]. Thus, derivatives of DFO, deferasirox (DFX) and deferiprone (DFP) have been developed and used for the treatment of iron overload in several countries. Even though both iron chelators can reduce iron levels in patients more effectively than DFO, unfortunately, these iron chelators have also some adverse effects on users, such as gastrointestinal upset, rash, renal toxicity, arthralgia, and hepatotoxicity [9]. Hence, the discovery of new iron chelators is needed for the treatment of iron overload.

The siderophore biosynthesis pathway is characterized in several fungi, including a pathogenic dimorphic fungus *Talaromyces marneffei* [10,11]. The biosynthesis is initiated by a precursor L-ornithine, a non-proteinogenic amino acid that plays a role in the urea cycle. The synthesis and assembly of siderophores are mediated by several enzymes and non-ribosomal cytoplasmic synthases. Regulation of iron homeostasis involves two major transcription factors, SreA and HapX [12,13]. They interconnectedly regulate iron uptakes in a negative feedback loop manner. In iron-limited conditions, the bZip-transcription factor, HapX, induces the siderophore biosynthesis pathway and represses iron consumption in the cells. In contrast, the GATA-factor, SreA, inhibits the iron acquisition process to prevent iron toxicity to the cells. Thus, the general idea to enhance the siderophore biosynthesis could be accomplished by either overexpressing the *hapX* or removing the *sreA*. A recent study in *T. marneffei* found that deletion of the *sreA* in *T. marneffei* led to increased levels of several transcripts of genes in the siderophore biosynthetic pathway [10]. Research on *Aspergillus fumigatus* also revealed that the absence of the *sreA* enhanced siderophore accumulation. However, this strain was susceptible to iron and oxidant toxicity; therefore, deletion of the *sreA* was unlikely to boost the fungal pathogenicity [13]. A similar result was found in *Aureobasidium pullulans*, where deletion of the *sre1* (*sreA*) gene enhanced the siderophore production in medium containing ferric iron (Fe^3+^). [14]. Recently, deletion of *sreA* in *Alternaria alternate* also led to the overproduction of siderophores [15]. Therefore, the *sreA* gene deletion approach was chosen in this study to generate a siderophore-enhancing strain in *T. marneffei*. Then, the *sreA* mutant was phenotypically characterized, especially at the amount of siderophore production. The produced siderophore was then purified and primarily tested for the ability to be used as an iron chelator, antimicrobial, or anticancer agent.

## 2. Materials and Methods

### 2.1. Chemicals and Reagents

Chrome azurol S (CAS) or trisodium 5-[(E)-(3-carboxy-5-methyl-4-oxocyclohexa-2,5-dien-1-ylidene)(2,6-dichloro-3-sulfonatophenyl)methyl]-3-methyl-2-oxidobenzoate (IUPAC name), dimethylsulfoxide (DMSO), fetal bovine serum (FBS), 3−[4,5−dimethylthiazol−2−yl]−2,5−diphenyltetrazolium bromide (MTT), and silica gel plates were purchased from Sigma-Aldrich Chemical Company, St. Louis, MO, USA. Dulbecco’s Modified Eagle Medium (DMEM) and RPMI1640 medium were obtained from Gibco. Calcein-acetomethoxy (Calcein-AM) was purchased from AAT Bioquest Inc., Sunnyvale, CA, USA.

### 2.2. Fungal Strains and Culture Conditions

*Talaromyces marneffei* wild-type ATCC18224 (FRR2161) and uracil auxotroph G816 strain (Δ*ligD niaD*^−^ *pyrG*^−^) were used in this study [16,17]. *T. marneffei* G816 strain was cultured on *Aspergillus* minimal medium (ANM) supplemented with 5 mM uracil and 10 mM (NH_4_)_2_SO_4_. *T. marneffei* FRR2161 and ∆*sreA* mutant strain (∆*sreA*, *pyrG*^+^) were cultured on the ANM without uracil. The conidia were harvested from a ten-day-old culture by scraping the colony surface and resuspending them in sterile normal saline-tween solution (0.1% *v*/*v* tween 40, 0.85% *w*/*v* NaCl). The suspensions were filtered through a sterile glass wool to remove the mycelia. The conidia concentration was enumerated by counting in a hemacytometer.

To study the growth of *T. marneffei*, 5 µL of the conidial suspension containing 5 × 10^4^ conidia of each strain was dropped onto the surface of the ANM medium. The colony was observed and measured the colony diameter after incubation at 25 °C for 3, 5, 7, and 12 days.

### 2.3. Generation of sreA Mutant Strain

*T. marneffei sreA* gene with approximately 1.5–2 kb of 5′ and 3′ flanking untranslated sequences were amplified from FRR2161 genomic DNA using primer *sreA*-*apa*I(F) (5′-TAATTGGGCCCTGCTTGG-3′) and *sreA*-*sal*I(R) (5′-TAAGTCGACTTATCGAGGTCGTGGCCTGA-3′). The PCR product was cloned into a pGEM-T vector (Promega Inc.), generating a plasmid, pAA_*sreA*. Deletion construct was generated by performing inverse PCR on plasmids pAA_*sreA* (primers inverse-*sreA*-*sma* I(F) (5′-TAACCCGGGTGAGCAAGAGGGCTGAATCT-3′) and inverse-*sreA*-*pst*I(R) (5′-TAACTGCAGGTTTCACTACGGCATGGGTT-3′) to remove coding region of the *sreA* gene. The *Aspergillus nidulans pyrG* (*AnpyrG*) selectable marker gene was removed from a pAB4626 plasmid [16] using *Sma* I and *Pst* I, and then ligated into the inverse PCR product to generate a deletion plasmid, pAD_*sreA*. The deletion construct was amplified from the deletion plasmid and transformed into protoplasts of *T. marneffei* G816 strain ∆*ligD*, by using a PEG/CaCl_2_ method as described previously [16,17]. The colonies of transformants were confirmed for the desired genotype, *sreA-* and *pyrG*^+^, by PCR and RT-PCR methods. First, PCR screening of all transformants was performed to check the integration of the deletion construct at the homologous site using primers Check_∆*sreA* (5′-GGACGACTGGTTAATGAGGTT-3′) and L29 (5′-GGACTTTGAGTGTGAGTGGAA-3′). The absence of *sreA* gene was detected using primers *sreA*-F (5′-CTCGGCTAGTTCGCGTTC-3′) and *sreA*-R (5′-GTTTGGAAGACCAGGGCATC-3′). The *sreA* null mutant was selected and named ∆*sreA*. Second, RT-PCR was performed on the total RNA isolated from the ∆*sreA* to confirm the absence of *sreA* transcript. Two micrograms of the total RNA were converted into complementary DNA by using a ReverTra Ace^®^ qPCR RT Master Mix (TOYOBO, Osaka, Japan). Primers used for amplification of the *sreA* transcript were Real_*sreA*-F (5′-CGTACCCTGCACCTCTAGCT-3′) and Real_*sreA*-R (5′-TCTCGTTGAAGCTGCACTCG-3′). An actin gene was used as an endogenous control (primer sequences; Act1F, 5′-TGATGAGGCACAGTCTAAGC-3′, and Act1R, 5′-CTTCTCTCTGTTGGACTTGG-3′).

### 2.4. Germination Assay

To determine the conidial germination, 10^8^ conidia/mL of *T. marneffei* FRR2161 and ∆ *sreA* strains were inoculated into ANM broth and incubated at 25 °C with continuous shaking at 150 rpm. The germlings were observed and counted under a light microscope at 3, 6, 9, 12, and 24 h after inoculation. The percentage of germinated cells (in 1000 counted cells) was determined at each time point.

### 2.5. Chrome Azurol S (CAS) Assay

The CAS colorimetric method was used to roughly quantify the amount of siderophore production [18]. One million conidia of each strain were inoculated into the ANM broth and cultured at 25 °C for three days. The cells and culture supernatant were separated by centrifugation at 10,000 rpm for 20 min. The cells were mechanically broken with 0.1-mm glass beads in a mini bead beater (Biospec, Bartlesville, OK, USA). Total protein concentration was determined in the cell lysate and the culture supernatant using the Bradford assay. The siderophore activity was determined in 2 mg of total protein. Briefly, CAS reagent (0.15 mM CAS, 0.015 mM FeCl_3_, 1.5 mM hexadecyl trimethyl ammonium bromide (HDTMA), and 1 M piperazine (pH 5.6)) was added to the protein mixture, and the absorbance was read at 630 nm (BioTek™ Synergy™ H4 Hybrid Microplate Reader, Winooski, VT, USA). An EDTA (Sigma-Aldrich Chemical Company, St. Louis, MO, USA) was used to generate a standard curve. Using this standard curve, the siderophore production from each fungal strain was calculated in terms of micromoles per gram of total protein. The percentage of the siderophore unit was calculated using the following formula: % Siderophore units = [(Ar − As)/Ar], where Ar = Reference absorbance at 630 nm, As = Absorbance of the sample at 630 nm.

### 2.6. Purification of the Extracellular Siderophore

A culture supernatant of ∆*sreA* containing extracellular siderophores was subjected to purification. Five liters of culture were prepared by inoculating the ANM broth with ∆*sreA* conidia at a final concentration of 1 × 10^6^ conidia/mL, then shaken at 200 rpm at 25 °C for seven days. After centrifugation, the culture supernatant was harvested and filtrated through a 0.45 µm filtered membrane. The supernatant was acidified to pH 2.0 with 6 M HCl and loaded on to deionized water (DW) equilibrate Amberlite XAD-2 column (20 × 5.0cm, Merck KGaA, Darmstadt, Germany). The acid supernatant was loaded into the column and subsequently washed with two bed volumes of DW. The bound siderophores were eluted from the column with methanol. Fifty fractions were collected and tested for the presence of a siderophore using CAS assay.

Fractions containing siderophores were pooled and concentrated in a rotary evaporator (Heidolph Hei-VAP precision, Berlin, Germany). The lyophilized sample was re-dissolved in 5 mL methanol and loaded into a Sephadex LH-20 column (50 × 1.5 cm, Sigma-Aldrich Chemical Company, St. Louis, MO, USA) to purify the siderophores based on their hydrophobicity and molecular weight. The filtrates were collected. Siderophores were analyzed by a thin layer chromatography (TLC) assay. Positive fractions were assayed by a Reverse-Phase High Performance Liquid Chromatography (RP-HPLC) (4.6 mm × 100 mm, Agilent zorbax SB-C18 column, Agilent Technologies Inc., Santa Clara, CA, USA), using solvent A (99.9% DW + 0.1% formic acid) and solvent B (99.9% acetonitrile + 0.1% formic acid) as mobile phases. Then, the purified fractions were pooled and evaporated. The sample was dissolved in DW, lyophilized, and stored at −20 °C.

### 2.7. Characterization of the Purified Siderophore

***Thin-Layer Chromatography Analysis***. The presence of siderophores in the fractions produced by the purification process was analyzed using silica gel plates, run in a solvent system containing *n*-butanol, acetic acid, and DW in a 12:3:5 ratio [18,19]. The plates were developed with an iron-detecting reagent (0.1 M FeCl_3_ in 0.1 N HCl). The formation of a wine-colored product indicates the presence of a hydroxamate-type siderophore and a dark-grey spot indicates the presence of a catechol-type siderophore.

***High-Performance Liquid Chromatography-Diode Array Detection (HPLC-DAD) Analysis.*** The purified siderophores were confirmed by RP-HPLC. Agilent column (4.6 mm × 100 mm, Agilent zorbax SB-C18 column, Agilent Technologies Inc., Santa Clara, CA, USA) was used and the mobile phase was composed of 0.1% formic acid in water (A) and 0.1% formic acid in acetonitrile (B) (time = 0 min, 100% A and 0% B; time = 20 min, 10% A and 90% B; time = 25 min, 100% A and 0% B; total run time 30 min). The injection volume was 10 µL, the flow rate was 0.5 mL/min, the column temperature was 40 °C, and the diode-array detector (DAD) wavelength was fixed at 254 and 435 nm. The purified siderophore was saturated by Fe^3+^ and detected at 435 nm.

***High-performance liquid chromatography–mass spectrometry (HPLC-MS) Analysis.*** A molecular mass of extracellular siderophore produced from *T. marneffei* ∆*sreA* strain was determined using HPLC-MS method [20]. Symmetry^®^ C18-type column (4.6 mm × 100 mm C18 column, 5 µm, Waters Corporation, Milford, MA, USA) was used and the mobile phase was composed of 0.1% formic acid in water (A) and 0.1% formic acid in acetonitrile (B) (time = 0 min, 98% A and 2% B; time = 11 min, 5% A and 95% B; time = 11.1–15 min, 98% A and 2% B; total run time 30 min). The injection volume was 5 µL, the flow rate was 0.3 mL/min, and the column temperature was set at 27 °C. The ionized compound was injected directly into Agilent 6490 Triple Quadrupole mass spectrometer (Model 6490 Triple Quad LC/MS, Agilent Technologies Inc., Santa Clara, CA, USA) performed in the positive electrospray ionization (ESI+) mode. The ions within are separated according to their mass per charge ratio (*m*/*z*). The ions then progress via a detector that calculates the molecular weight of irons/fragments.

***Fourier-Transform-Infrared (FTIR) Spectroscopy Analysis.*** The FTIR method was used to identify the siderophore produced by *T. marneffei* isolate. Lyophilized siderophore sample was pelleted with potassium bromide (KBr) and subjected to FTIR spectroscopy (Bruker Tensor 27 FT-IR Spectrometer, Bruker Optics, Bruker Daltonics, GmbH and Co., KG, Breman, Germany) for a determination of functional groups. Spectra were recorded in the range from 4000 to 400 cm^−1^.

***NMR Analysis.*** To confirm the extracellular siderophore structure of *T. marneffei*, a nuclear magnetic resonance method was employed. The structure of the purified siderophore was analyzed using the ^1^H- and ^13^C-NMR technique. The siderophores were dissolved in 0.5 mL of deuterium oxide (D_2_O). ^1^H- and ^13^C-NMR spectra were recorded using an internal deuterium lock at ambient probe temperatures on a Bruker NEOTM 500 MHz NMR instrument (AVNACE NEO500 Ascend Bruker BioSpin International AG, Fallenden, Switzerland). Chemical shifts (δ) are quoted in part per million (ppm), to the nearest 0.01 ppm (for ^1^H NMRs) or 0.1 ppm (for ^13^C NMRs), and are referenced to the residual non-deuterated solvent peak. An interpreted spectrum of data was compared with the literature.

### 2.8. Cytotoxicity

To investigate the cytotoxic effect of isolated siderophore from *T. marneffei* on Huh7 cells, the viability of the cell was determined using a colorimetric MTT method [19]. The Huh7 cells were cultured in DMEM supplemented with 10% FBS, penicillin (100 IU/mL), and streptomycin (100 µg/mL) at 37 °C in a humidified atmosphere (5% CO_2_) and harvested at 80–90% confluence. The harvested Huh7 cells were seeded in 96-well culture plates, treated with various concentrations of the purified siderophores (final concentrations of 0–200 µg) for 24 and 48 h, washed with phosphate-buffered saline (PBS) pH 7.0, and subsequently incubated with MTT (5 mg/mL, Sigma-Aldrich Chemical Company, Poole, UK) for four hours. Finally, the blue formazan product was solubilized with DMSO (0.1 mL), and the absorbance was measured at 570 nm. The viability of the treated cells is presented in comparison with untreated cells (100% viability).

### 2.9. Labile Iron Pool (LIP) Assay

The cytosolic labile iron pool (LIP) consists of redox-active non-heme iron, which can be scavenged by iron chelators and detected by using the calcein quenching method [21]. Briefly, Huh7 cells were seeded in 96-well culture plates (1 × 10^4^ cells/well), loaded with 0.1 mM ferrous ammonium citrate (FAC) for two hours, and incubated with calcein-AM solution at 37 °C in the dark for 15 min and finally treated with various concentrations of the siderophore fractions for four hours. Fluorescence intensity (FI) was measured using a microplate spectrofluorometer with an excitation wavelength at 495 nm and emission wavelength at 515 nm. The percentage of the Fluorescence intensity of the sample was calculated using the following formula: % Fluorescence intensity (%FI) = (FI sample)/(FI control) × 100, where FI sample = Fluorescence intensity of sample at 495/515 nm, FI control = Fluorescence intensity of control at 495/515 nm.

### 2.10. Antimicrobial Activity Testing by Broth Microdilution Assay

*Staphylococcus aureus* (ATCC 25923), *Escherichia coli* (ATCC 25922), and *Candida albicans* (ATCC 90028) were used as the representatives of Gram-positive bacteria, Gram-negative bacteria, and fungi, respectively. To prepare the initial inoculum, each bacterium was cultured in a Mueller Hinton broth (MHB) at 37 °C for 4 h with continuous shaking. Then, the bacterial amount was adjusted to OD_600_ nm of 0.8 (equivalent cell number, 1 × 10^8^ cells/mL). To prepare a fungal inoculum, *Candida albicans* was cultured in 37 °C Sabouraud’s dextrose broth for 24 h. The yeast cells were enumerated and adjusted to 2.5 × 10^4^ cells/mL in a RPMI 1640 medium.

Antimicrobial activity testing used 50 µL of the bacterial suspension and 100 µL of the yeast suspension. Either 50 µL or 100 µL of coprogen B solution was added to the tested bacterial cells or yeast cells to final concentrations of 0, 6.25, 12.5, 25, 50, and 100 µg/mL. DFO at the same final concentration as coprogen B was used as the control. The plate was incubated at 37 °C for 18 h for *S. aureus* and *E. coli*, and 24 h for *C. albicans*. After that, 0.068% (*w*/*v*, final concentration) resazurin dye (Sigma-Aldrich, MO, USA) was added to each well and incubated at 37 °C for 24 h. A microplate reader (absorbances at 570 and 600 nm) was used to measure the growth of bacteria and yeast. A percent reduction of resazurin was calculated. The percentage of growth and inhibition was calculated as described previously [22].

### 2.11. Statistical Analysis

The data were analyzed, depending on the experiment, with a Student’s *t*-test and Tukey’s multiple comparison test or unpaired *t*-test and Welch’s correction with a significant value of *p* < 0.05. All statistical analysis was performed by using Prism software (GraphPad, version 7.0).

## 3. Results

### 3.1. Characteristics of Talaromyces Marneffei ∆sreA Strain

The effects of *sreA* gene deletion on *T. marneffei* colony morphology and growth rate were examined. *T. marneffei* ∆*sreA* strain and FRR2161 wild-type strains were grown on the ANM minimal medium (low iron medium, 7 µM). A normal characteristic of *T. marneffei* wild-type strain showed a velvety to fluffy colony appearance. The formation of conidia is visualized as green at the center, while hyphae are made white at the periphery of the colony in Figure 1A. The colony of the mutant showed slightly different characteristics. The colony was denser and more compact compared to the wild type (Figure 1B). The colony diameters were reduced in the mutant grown on ANM when compared to the wild type (Figure 1C), indicating the growth reduction. The germination rate was determined by counting the germlings and germinated conidia in ANM broth after cultivation at 25 °C for several time points. The germination of both strains began at the same time, at six hours after incubation. However, the Δ*sreA* showed lower numbers of germinated cells compared to the wild type (Figure 1D). This result revealed that deletion of the *sreA* gene affected the conidial viability. However, cultivation of the mutant on a standard mycological medium, such as potato dextrose agar (PDA), did not show the growth reduction. This result implied that iron availability in the medium is important for the growth of ∆*sreA.*

### 3.2. Deletion of sreA Increased the Synthesis of Siderophores

The ability of the ∆*sreA* strain to produce siderophores was assessed by Chrome Azurol S (CAS) assay. *T. marneffei* wild type and ∆*sreA* strains were cultured in ANM broth for seven days, centrifuged to separate the cells and culture supernatants. The total siderophore produced and accumulated inside the cells (intracellular siderophores) and secreted into the culture supernatant (extracellular siderophores) were initially determined by a CAS agar plate method. A circular membrane was impregnated with each fraction, dried and laid down on the surface of the CAS agar plate. After incubation, a halo formation around the membrane indicates the presence of siderophores. From the result, Δ*sreA* showed an increased siderophore production, both intracellular and extracellular compared to the wild type (Figure 2A). Quantification of siderophores by CAS liquid assay found significantly elevated extracellular amounts of siderophores in ∆*sreA* (Figure 2B). Siderophores were then purified from the culture supernatant by using Amberlite XAD-2 and Sephadex LH-20 columns. Amberlite’s collected fractions were analyzed with the CAS agar plate and found the orange halo zone from fractions 4–7 (4 out of 8 fractions collected), while only one fraction was found positive in the wild type. The positive fractions from XAD-2 were then purified by Sephadex LH-20. TLC analysis showed the presence of iron complexing compounds in 10 fractions (F24-F33). All fractions contained only one spot at a retention factor of 0.635, indicating the presence of a single form of siderophore (Figure 2C). Therefore, all fractions were pooled and concentrated. The total amount of siderophores, as concentrated into a lyophilized powder, was approximately 19.43 times higher than those isolated from the wild type (Figure 2D).

### 3.3. Characterization of Secreted Siderophore of ΔsreA by RP-HPLC, LC-MS, FTIR, and NMR

To investigate types of siderophore purified from *T. marneffei* Δ*sreA* strain. All of the siderophore-positive fractions were pooled and analyzed by RP-HPLC. The results were compared to published data [23] to identify the secreted siderophore type. The predominant peak was identified as a coprogen B (retention time at 4.806 min) (Figure 3).

The molecular weight of the purified extracellular siderophore from Δ*sreA* mutant strain was verified by LC-MS method. The result of mass spectrometric analysis yielded a molecular mass of 726.8000 *m*/*z* [M+H]^+^, perfectly matching coprogen B when compared to the database (https://pubchem.ncbi.nlm.nih.gov/compound/Coprogen-B, accessed on 23 August 2022). This finding suggested that it has a molecular weight of 726.8000 (Figure 4).

Additionally, FTIR analysis was performed to investigate the functional group of hydroxamate siderophore from the purified siderophore. The IR spectrum data showed the presence of O-H stretching at the 3550–3200 cm^−1^ region, indicating the presence of a primary alcoholic group. This study’s findings are consistent with those of Patel et al. (2009) [24] and Murugappan et al. (2011) [25], who identified a nearby hydroxyl peak at 3357.21 cm^−1^ [24,25]. The second peak between 2927 and 2938 cm^−1^ refers to the CH aliphatic group. In addition, the amide (C=O) and amine (N-H) groups were identified by the absorption peaks in 1745 and 1651 cm^−1^ (Figure 5).

The purified siderophore was analyzed with NMR spectroscopy to confirm its structure. ^1^H-NMR and ^13^C-NMR spectra of an unbound form of siderophore were measured in D_2_O. The ^1^H-NMR spectra showed presence of δ 6.25 (3H, brs, 7/7′/7’’-H), 4.30–4.40 (6H, m, 5/5′/5”-H2), 4.08 (6H, t, *J* = 6.28 Hz, 10/10′/10’’-H2), 3.70 (3H, m, 2/2’/2’’-H), 2.80 (6H, m, 9/9′/9’’-H2), 2.60 (2H, m, 9’-H2), 1.86 (9H, s, 11/11’/11’’-H3), and 1.7–1.6 (12H, m, 3/3’/3’’-H2, 4/4’/4’’-H2) (Figure 6, Table 1). ^13^C-NMR spectrum revealed presence of δ 169.9 (C-1’/C-1’’), 168.4 (C-1), 168.4 (C-6/6’/6’’), 151.3 (C-8/8′/8’’), 117.4 (C-7/7′/7”), 64.8 (C-10/10′/10’’), 52.4 (C-2/2’/2’’), 46.8 (C-5/5’/5’’), 43.3 (C-9/9′/9’’), 27.9 (C-3/3’/3’’), 23.7 (C-4/4’/4’’), and 21.9 (C-11/11’/11’’) (Figure 6, Table 1).

The proton and carbon NMR resonances for this siderophore were compared to the known data for the coprogen family and the results of these NMR spectra are consistent with Huang et al. (2020) [26,27,28,29]. The results of spectral data corresponded extremely well with those shown in the literature, confirming the secreted siderophore from *T. marneffei* ∆*sreA* as coprogen B (Figure 7).

### 3.4. Testing of Cytotoxic Effect and Iron Chelating Property of Coprogen B

#### 3.4.1. Effect of Coprogen B on Cell Viability

Cytotoxic effect of coprogen B isolated from *T. marneffei* ∆*sreA* strain was investigated in a Huh7 hepatocarcinoma cell line. At 24 h after treatment, both the coprogen B and DFO showed no toxicity to the Huh7 cell (viability more than 90%) (Figure 8A). However, the DFO showed concentration-dependent cytotoxicity at 48 h of treatment (viability less than 50%), while the purified coprogen B showed no toxicity (viability more than 78%) (Figure 8B). Since the human cancer cell is used in the assay, we concluded that the purified coprogen B had no anticancer activity. However, since it showed no toxicity to the human cell, it can also be safely applied for human uses.

#### 3.4.2. Potential of Cellular Iron-Chelating Activity

To investigate an iron-chelating activity, purified coprogen B was tested with an iron-loaded cell model. In the experiment, the Huh7 cells were loaded with iron (from ferric ammonium citrate); thus, the LIP was significantly increased. The ability of the iron chelators to chelate and remove LIP from the cells was measured by a calcein quenching method [21,30]. In the presence of the iron chelators, an elevation of fluorescence intensity (FI) derived from the calcein indicates a decrease of irons inside the cells (labile iron pool, LIP). Both the coprogen B and deferoxamine could decrease the LIP levels in a dose-dependent manner, and a significant reduction was found at 1 milligram (Figure 9 and Figure 10A,B). However, at the same amounts of chelators used, coprogen B could reduce LIP levels more than DFO (Figure 10C). When comparing the compounds at the equivalent concentrations, the chelation efficiency of coprogen B was higher than deferoxamine as demonstrated by the ability to reduce LIP (∆FI of coprogen B vs DFO at 12.5, 25, and 50 µg were 19.31, 19.39, and 16.33, respectively) (Figure 10C). Therefore, our results suggested that the coprogen B derived from *T. marneffei* could be used as an alternative iron chelator drug in the future.

### 3.5. Evaluation of Ability of Coprogen B to Inhibit Growth of Escherichia coli, Staphylococcus aureus, and Candida albicans

Deferoxamine is a standard iron chelator, used to treat iron overload patients [8]. It has been observed that it was able to promote systemic bacterial infections, thus harming patients [31]. We aim to investigate whether purified coprogen B has a similar effect. A standard broth microdilution assay for antibacterial and antifungal susceptibility tests was used. First, 1 × 10^6^ cells/well of both bacteria and 250 cells/well of yeast were treated with various concentrations of either the purified coprogen B or DFO (0, 6.25, 12.5, 25, 50, 100 µg/mL, and 1 mg/mL). Then, the percentage of growth and inhibition was calculated. The results found that both coprogen B and DFO could inhibit the growth of *C. albicans*, but coprogen B is more effective than DFO, especially at 50–1000 µg/mL (Figure 11A). Importantly, the coprogen B did not enhance the growth of *C. albicans*, implying that this compound can be useful as an iron chelator by not triggering the overgrowth of this opportunistic fungus. In bacterial cases, the high concentration of coprogen B could inhibit both *E. coli* and *S. aureus* growth, whereas low concentrations could promote their growth (Figure 11B,C). The same effect was observed in DFO. Therefore, the optimal concentration is needed to be determined if the coprogen B would be used therapeutically. From this preliminary result, we concluded that coprogen B has both antimicrobial activity and supporting activity depending on its concentration and types of pathogen.

## 4. Discussion

All microorganisms require iron to maintain cellular function and growth [32]. They have evolved several mechanisms to scavenge and absorb iron from the environments. Siderophore production is one of the most common mechanisms for acquiring iron. The siderophores are low molecular weight compounds (500–1500 Da) with a high affinity and selectivity for iron (III) [33]. The bacterial-derived siderophores have been developed as iron chelators in clinical and applications and biotechnology. Even though the fungi produced more diverse siderophores with high iron binding efficiency, the fungal siderophores have got less attention and applications due to their small production. Unlike bacteria, fungi grow slowly that take several days to weeks, and they require a large volume of medium and aeration. Although, the modification of the culture medium without iron allows the fungi to produce an increased siderophore level, it is still not enough for purification and application processes. In this study, we successfully applied the genetic engineering of *Talaromyces marneffei* to generate an enhanced siderophore-producing strain. The siderophores production has been increased nearly 20 times compared to the wild type.

Control of iron metabolism is tightly regulated in fungi since iron is a vital element [11]. Normally, the SreA is required to protect fungal cells from iron cytotoxicity by limiting the iron uptake mechanisms. Therefore, deletion of the *sreA* approach could be harmful to some organisms. Our study found that the ∆*sreA* strain had slightly reduced growth when cultured on the iron-limited ANM medium. However, the growth rate was recovered when the organism was grown on the iron-sufficient PDA medium (approximately 20 µM). Similarly, deletion of the *sreA* homolog in other fungi; *Aspergillus fumigatus*, *A. nidulans*, *Aureobasidium pullulans*, *Ustilago maydis*, and *Shizosaccharomyces pombe* resulted in little or no growth reduction [13,14,34,35,36,37]. In contrast, ∆*sreA* mutant in *Alternaria alternata* showed reduced growth both on minimal medium and PDA, but the growth reduction was more prominent on PDA [15]. Therefore, the concentration of irons in the medium is needed to be optimized to determine an optimal condition for the highest growth and siderophore production in ∆*sreA* strains. Despite the mutation of *sreA* in *T. marneffei* having some effect on the mycelial growth in ANM (containing low iron, 7 µM), this condition promotes fungal siderophore production. The ANM medium was, thus, chosen as the culture medium for this research. *T. marneffei* ∆*sreA* strain could produce a high amount of coprogen B, a type of extracellular siderophores that can be purified from the culture filtrate in normal culture condition. The extracellular siderophore yield obtained in this work was extremely high, which was 200 mg of siderophores isolated from a liter of culture. The optimal condition, which is yet to be determined, thus, should have provided higher yields in the future.

The present research confirmed that *T. marneffei* produces the coprogen B hydroxamate type of siderophore and excretes it into the culture supernatant. This result was expected since the same result was obtained from Pasricha’s study. The siderophore synthesis pathway in *T. marneffei* suggested that coprogen B is produced from the function of *sidD,* which encodes one of the non-ribosomal peptide synthases (NRPS) [10]. This orthologous gene, *sidD,* was found in *A. fumigatus*, but it plays a role in fusarinine C production instead of coprogen B [11,12]. Since the encoded product from *sidD* gene of *T. marneffei* has never been characterized, the catalytic function of TmSidD is unknown. However, an ascomycetous fungal coprogen synthetase Nps6, has been described [38]. The molecule consists of the core domain for adenylation, thiolation, and condensation. The adenylation domain (A-domain) of Nps6 recognizes and activates a *N^5^*-*trans*-anhydromevalonyl-*N^5^*-hydroxy-*L*-ornithine (*trans*-AMHO) as its acyl-adenylate by reaction with ATP. Then, the thiolation domain (T-domain) as a peptidyl carrier protein (PCP) domain utilizes a terminal thiol of a post-translationally installed phosphopantetheinyl (PPant) arm to bind the activated carboxyl group of adenylate. The condensation domain (C-domain) then catalyzes directly to transfer another intermediate of acyl amino acid in the adjacent posterior form to generate peptide bond. The C-domain also catalyzes intra-molecular cyclization to yield the diketopiperazine moiety resulting in the release of coprogen product from NRPS [38,39,40,41]. The TmSidD is expected to share the Nps6 structure to perform the same function.

The structure of coprogen B had been previously characterized in non-human pathogenic fungi [28,42]. The result of this study confirmed the same chemical structure of *T. marneffei’s* coprogen B with others. Chemical characterization identified the purified siderophore from the ∆*sreA* as the coprogen B by LC-MS, FTIR, and ^1^H and ^13^C NMR methods (Table 1). This same structure had also been reported in other medically important Ascomycetous filamentous fungi, such as *Penicillium chrysogenum*, *Alternaria alternata*, *Aspergillus niger*, and *Histoplasma capsulatum* [43,44,45,46,47,48]. The affinity constants for iron (III) of the coprogen family had been proven by EDTA competition reactions yielded the values log K* = 4.6 and logKFeIII of approximately 30.2, which is close to deferoxamine [49]. Additionally, it has been reported that the coprogen family has a higher affinity for iron (III) than other hydroxamates and transferrin [49]. Therefore, the ability of coprogen B to bind, chelate, and form a complex with iron (III) suggested that coprogen B has the potential to be used as an alternative iron chelator.

Deferoxamine (DFO) is used as the iron-chelating drug for iron-overloaded patients. This siderophore is derived from a bacterium; *Streptomyces pilosus*. Deferoxamine (DFO) and coprogen B are the hexadentate trihydroxamate siderophores. The iron-binding affinity constants of both siderophores are equivalent [49,50]. However, the DFO possessed some side effects in therapeutic uses. Interestingly, the purified coprogen B in this study showed no toxicity to the human Huh7 cell line when using concentrations up to 100 µg, whereas the DFO was found to be toxic, even at 25 µg after 48 h of incubation. A similar result was obtained in another study using the coprogen siderophore produced from *Penicillium nalgiovense*. It showed no cytotoxic effect on the HaCat human keratinocytic cell line [51,52]. Moreover, the ability of coprogen B isolated from *Penicillium chrysogenum* was proven to inhibit the heme-catalyzed oxidation of the low-density lipoprotein (LDL), resulting in a reduction of vascular endothelium damage and atherosclerosis [51,52,53]. Iron chelator has been verified that it can remove and chelate iron from the LIP and limit its involvement in ROS production [54]. The reduced LIP levels might contribute to the reduction of the production of oxygen radicals that cause damage to cellular biomolecules [55,56,57,58]. However, the iron-chelating activity of coprogen B has never been investigated. Our study has demonstrated the ability of coprogen B to effectively eliminate the excess iron from the iron-loaded cell in a dose-dependent manner. Our findings demonstrated that *T. marneffei*’s coprogen B has the potential to be an iron since it could withdraw irons from the iron-loaded cells without disturbing the viability of the human cell.

Some microorganisms develop a xenosiderophore uptake mechanism to steal irons from surrounding microflora, and *Candida albicans* is one of them. *C. albicans* contributes to the gut microflora, which can also cause opportunistic infections. After the iron chelator binds the irons from the LIP, it had to be excreted into the gastrointestinal tract or urinary system; therefore, it is important to ensure that the ideal iron chelator to be used in therapeutics would not be taken by this opportunistic pathogen. As previously known, the deferoxamine drug has been reported to play dual roles in the growth promotion of the pathogen and immunosuppression, which negatively affected patients with iron overload [59]. We performed preliminary antimicrobial testing to determine the ability of the purified coprogen B and found that it did not enhance the growth of *Candida albicans*. In contrast, it possessed antimicrobial activity to *Escherichia coli* and *Staphylococcus aureus* dose-dependently. This antibacterial effect of iron chelators is not surprising, since other candidates gained similar results. For example, VK28 (Varinel, Inc., West Chester, PA, USA) could inhibit the growth of *Acinetobacter baumannii*, *Escherichia coli*, and *Staphylococcus aureus*; ApoL1 and Apo6619, could also suppress the growth of some strains of *Pseudomonas aeruginosa, Klebsiella pneumoniae*, *A. baumannii,* and *E. coli* [60]. The only concern is that, at very low concentrations, the coprogen B could also promote bacterial growth. Therefore, this is one of the issues to be a concern in the future if this molecule will be used as an iron chelator.

A previous study revealed that synergistic interactions between amphotericin B and lactoferrin or amphotericin B and deferoxamine could enhance the antifungal activity of amphotericin B and minimize the dosage required to kill *Cryptococcus* spp. [61]. Moreover, the synergistic activity of siderophore and antibiotic could enhance the inhibition of methicillin-resistant *Staphylococcus aureus* (MRSA), metallo-β-lactamase-producing *P. aeruginosa*, and *A. baumannii* growth [62]. Thus, future research should investigate the synergistic effect of coprogen B and antimicrobials to minimize the therapeutic dose, boost efficacy, and prevent pathogen resistance. The ability of *T. marneffei* siderophore, coprogen B, to bind iron tightly suggests the possibility of a high affinity, which could be further developed into a siderophore-antimicrobial drug conjugate. A recent study demonstrated that an extracellular siderophore, triacetylfusarinine C (TAFC), of *A. fumigatus* with a variety of antifungal molecules conjugates revealed the ability to deliver (Trojan horse strategies) antifungal compounds specifically into *A. fumigatus* hyphae through the siderophore Iron Transporter (SIT) subfamily of the major facilitator protein superfamily to inhibit fungal growth [13,63,64]. Additionally, modified siderophores are also attached to fluorescent probes or radiolabeled with gallium-68 to allow molecular imaging as current diagnostic approaches. Previous studies have also revealed that gallium (III)-siderophores (Gallium-68 radiolabeling) can be used as a tool for imaging the location of *S. aureus* and *A. fumigatus* infection in mice models [3,64,65,66,67]. Thus, coprogen B might be further developed as a diagnostic marker that combines with gallium-68 to detect the location of *T. marneffei* infection using Positron Emission Tomography and Computed Tomography (PET/CT) technology.

## 5. Conclusions

*Talaromyces marneffei* ∆*sreA* mutant produced a large amount of the coprogen B extracellular siderophore. Identification of *T. marneffei*’s coprogen B found the same structure as previously reported from other Ascomycetous fungi. Coprogen B has an advantage as a potential iron chelator as shown in our preliminary assays. It could withdraw irons from the iron-loaded cells without disturbing the viability of the human hepatocytes. Importantly, the coprogen B did not support the growth of a gut-residing fungus *Candida albicans*, suggesting that it could be used safely in therapeutic purposes. It showed antimicrobial activity by inhibiting the growth of *Staphylococcus aureus*, *Escherichia coli,* and *Candida albicans* in a dose-dependent manner. Therefore, the coprogen B isolated from the ∆*sreA* siderophore-enhancing strain of *T. marneffei* could be useful in the development of a new iron-chelating agent and other medical applications.

## Figures and Tables

**Figure 1 jof-08-01183-f001:**
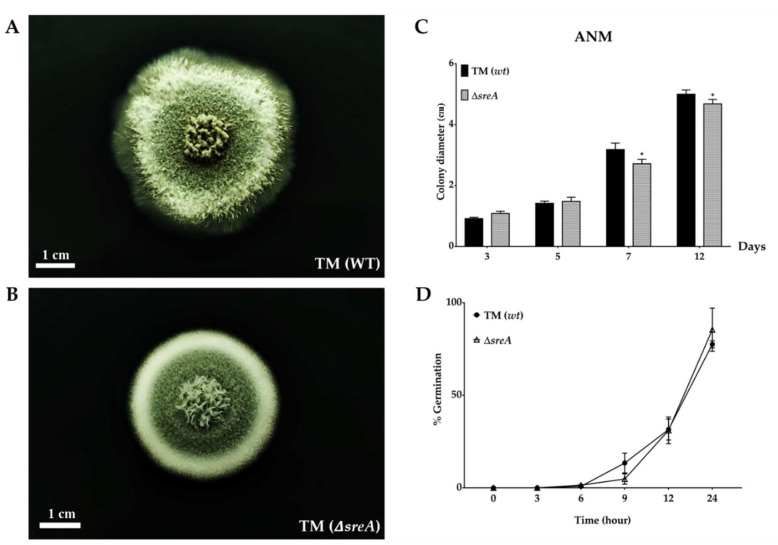
Colony growth and conidial germination of *T. marneffei* strains. *T. marneffei* strains were cultured on ANM agar at 25 °C and observed for up to 12 days. The typical green velvety colony is observed in the 7-day-old FRR2161 strain (**A**). A colony of ∆*sreA* (7 days) showed a similar appearance but was more compact (**B**). The colony diameters of the ∆*sreA* are significantly reduced compared to the wild type (*p* value < 0.01) (**C**). The percentage of germination is determined by one thousand total cell counts (**D**). Asterisk (*) indicates data that significantly differed based on Tukey’s test.

**Figure 2 jof-08-01183-f002:**
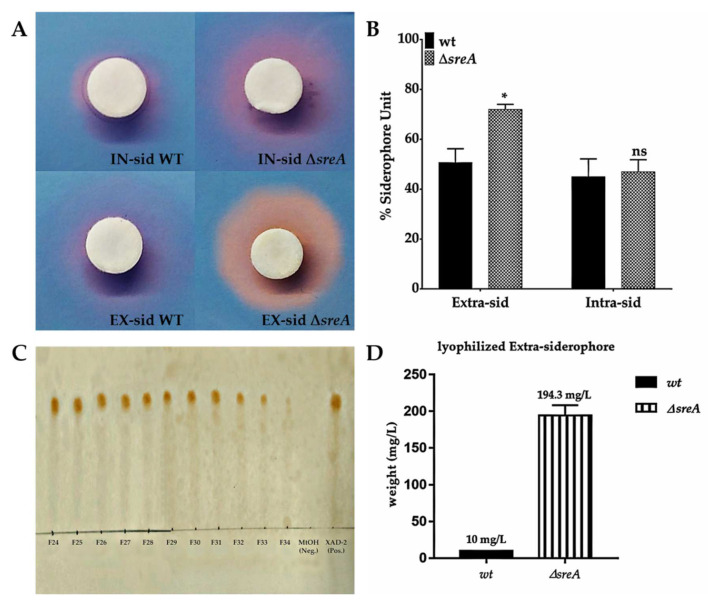
Detection of siderophores produced by *T. marneffei* strains. The halo zone of intracellular and extracellular siderophores was observed by Chrome azurol S (CAS) agar assay (**A**). Intracellular and extracellular siderophores levels were measured by CAS liquid assay (**B**). Thin-layer chromatographic analysis of purified fractions (**C**). The stationary phase was TLC silica gel 60 F_254_, the mobile phase was a solvent mixture of n-butanol: acetic acid: DI water (12:3:5*, v*/*v*/*v*). Developing agent was 0.1 M FeCl_3_ in 0.1 N HCl. The siderophore activity of Δ*sreA* mutant was compared with the *T. marneffei* wild type. The weight of the lyophilized compound was measured from *T. marneffei* ∆*sreA* strain compared with wild type (**D**). The significant levels were determined by the unpaired *t*-test. Asterisk (*) indicates the difference between wild type and ∆*sreA*, *p*-value <0.0001.

**Figure 3 jof-08-01183-f003:**
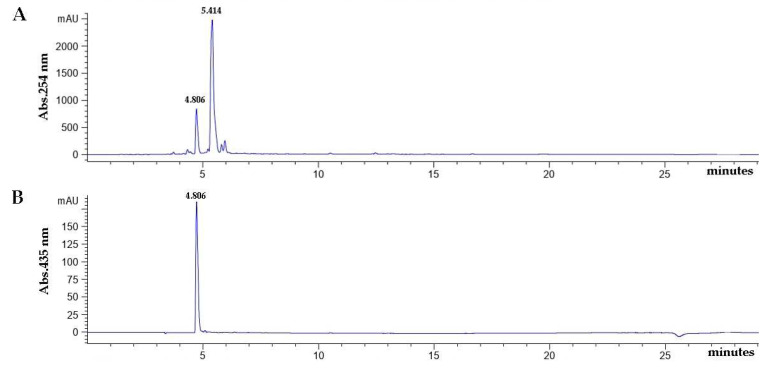
HPLC-visible chromatograms of the secreted siderophore extracts from the culture supernatant of *T. marneffei* ∆*sreA*. The partially purified siderophore was analyzed by reverse-phase HPLC using absorbance determination at 254 (**A**) and 435 nm (**B**). The peak at retention times of 4.806 and 5.414 min are the ferri- (iron-bound) and desferri-form coprogen B siderophore, respectively.

**Figure 4 jof-08-01183-f004:**
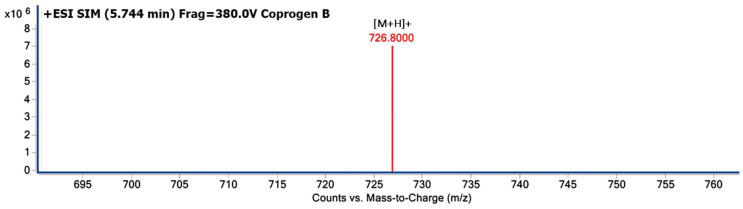
LC-MS chromatogram of the purified siderophore from the culture supernatant of *T. marneffei* ∆*sreA*. These chromatograms show the mass spectra for iron-free siderophores in positive ion mode. The major peak (red) indicates that the molecular weight of the compound is 726.8000 *m*/*z*, [M+H]^+^.

**Figure 5 jof-08-01183-f005:**
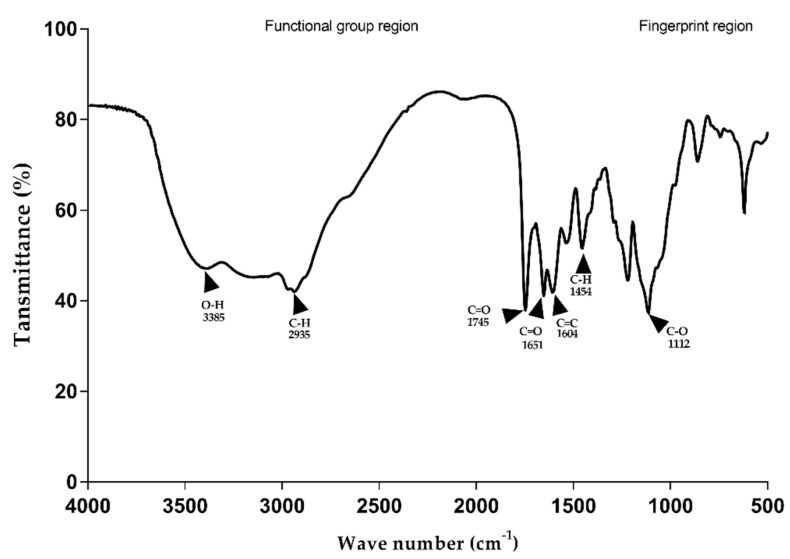
Investigation of the functional group of coprogen B from *T. marneffei* ∆*sreA* strain by FTIR. FTIR analysis of the siderophore shows the presence of hydroxyl group (O-H stretching at wavenumber 3550–3200 cm^−1^), ester group (C=O stretching at wavenumber 1745 cm^−1^), amide group (C=O bending at wavenumber 1651 cm^−1^), and alkene (C=C stretching at wavenumber 1604 cm^−1^).

**Figure 6 jof-08-01183-f006:**
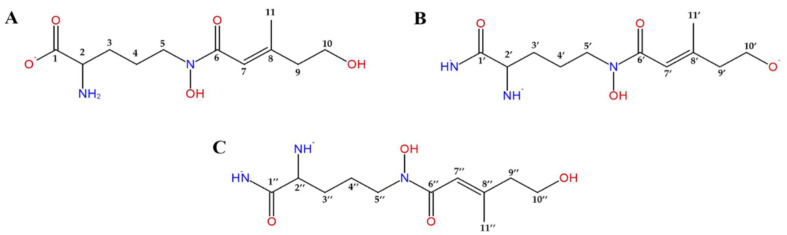
Three fragment structures of coprogen (**B**) are investigated by ^1^H-NMR and ^13^C-NMR spectroscopy. Fragment (**A**) is linked to fragment (**B**) via an ester group. Then, fragments (**B**) and (**C**) are linked through the amide group.

**Figure 7 jof-08-01183-f007:**
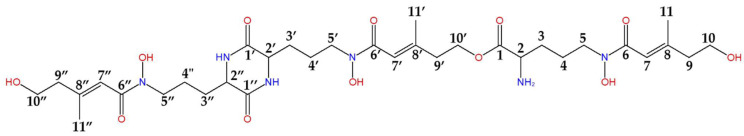
Coprogen B structure investigated by ^1^H-NMR and ^13^C-NMR spectroscopy.

**Figure 8 jof-08-01183-f008:**
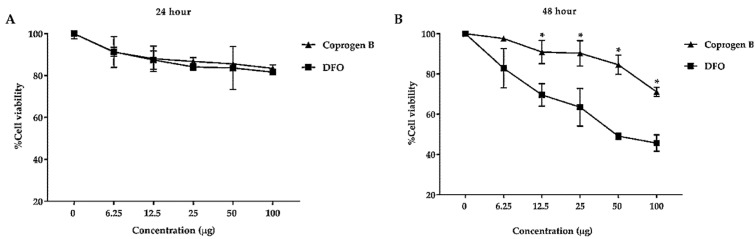
Investigation of the cytotoxic effect of coprogen B on human hepatocellular carcinoma cells. Viability of human hepatocellular carcinoma (Huh7) cells were treated with coprogen B (▲) and deferoxamine (DFO) (■) for 24 h (**A**) and 48 h (**B**). Data obtained from three independent triplicate experiments are expressed as mean ± SD. Asterisk (*) indicates significant difference *(p*-value < 0.0001) when compared between each concentration of coprogen B and DFO.

**Figure 9 jof-08-01183-f009:**
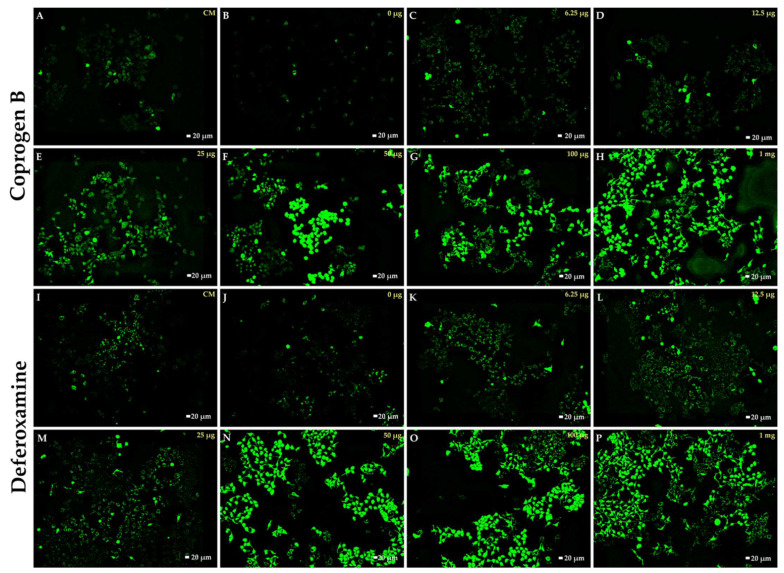
Effect of Coprogen B and deferoxamine on labile iron pool levels in iron-loaded human hepatocellular carcinoma (Huh7) cells. The ferric ammonium citrate was loaded into Huh7 cells to produce iron-loaded cells. The iron-loaded Huh7 cells were stained with 0.2 µM calcein-AM. After that the iron-loaded cells were treated with various concentration (0, 0.00625, 0.0125, 0.025, 0.05, 0.1, and 1 mg) of coprogen B (**B**–**H**) and deferoxamine (**J**–**P**). The green fluorescence intensity in the cytosol of Huh7 cells was observed by fluorescence microscopy (10×). The upper panel represents the cell treated with purified coprogen B. The lower panel represents the cell treated with deferoxamine. The CM (**A**,**I**) represents the cell did not load the iron and was treated either with coprogen B or deferoxamine.

**Figure 10 jof-08-01183-f010:**
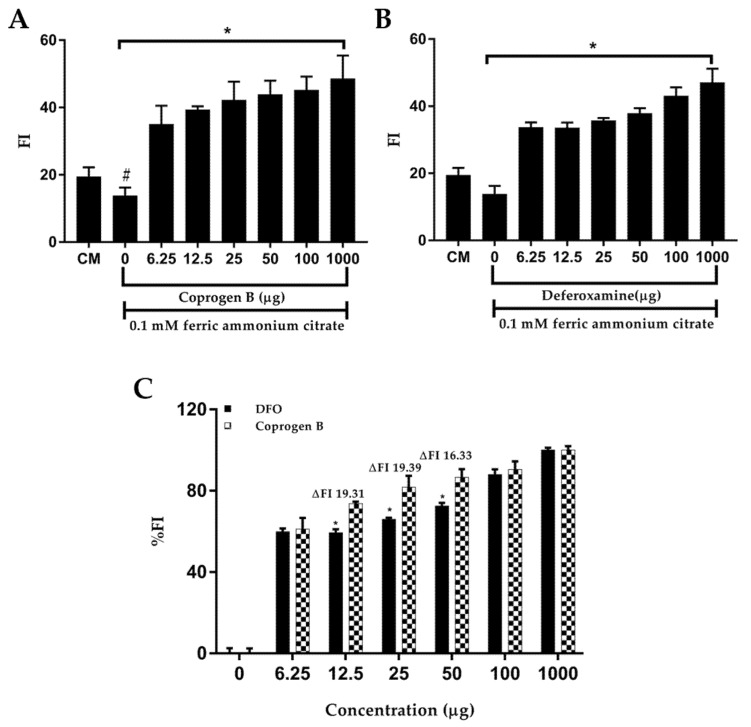
Evaluation of the ability of coprogen B and deferoxamine to reduce labile iron pool (LIP) levels in iron-loaded human hepatocellular carcinoma (Huh7) cells. The calcein fluorescent intensity was detected by a microplate reader (excitation wavelength at 495 nm and emission wavelength at 515 nm). Elevation of fluorescence intensity (FI) showing the reduction of the LIP is a consequence of the intracellular binding of ferric iron by coprogen B (**A**) and deferoxamine (DFO) (**B**). Comparison between percentage fluorescence intensity (%FI) of DFO or coprogen B treated iron overloaded Huh7 cells at various concentrations (**C**). The data show more effective binding of coprogen B than DFO at 12.5 to 50 μg. ∆FI is the different percentage fluorescent intensity. Data obtained from three independent triplicate experiments are expressed as mean ± SD, # *p* < 0.05 when compared with culture medium (control); * *p* < 0.05 when compared with ferric ammonium citrate (FAC) treatment alone.

**Figure 11 jof-08-01183-f011:**
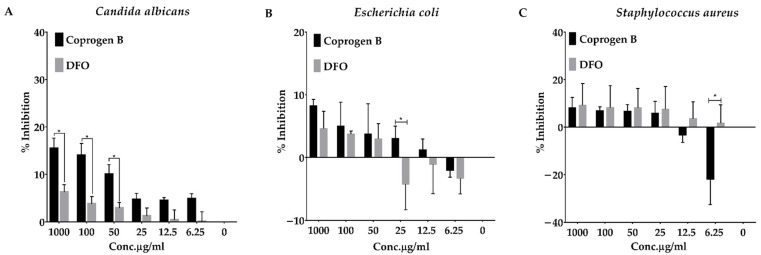
Effect of coprogen B and deferoxamine on growth of *Candida albicans* (**A**), *Escherichia coli* (**B**), and *Staphylococcus aureus* (**C**). The level of growth in each well was quantified by absorbance at 570 and 600 nm. Then, the percentage of inhibition was calculated from the percentage reduction of resazurin dye. Data are expressed as mean ± SD. Asterisk (*) indicates significant difference, *p*-value < 0.05.

**Table 1 jof-08-01183-t001:** ^1^H-NMR (500 MHz) and ^13^C NMR (125 MHz) chemical shifts of coprogen B in D_2_O.

Position	^1^H Chemical Shift δ in D_2_O (ppm)	^13^C Chemical Shift δ in D_2_O (ppm)	Structural Group
1		168.4	>C=O
1′/1”		169.9	>C=O
2/2′/2”	3.7, m	52.4	>CH-
3/3′/3”	1.6–1.7, m	27.4	-CH_2_-CH_2_-
4/4′/4”	1.6–1.7, m	23.7	-CH_2_-CH_2_-
5/5′/5”	4.3–4.4, m	46.8	-CH_2_-CH_2_-
6/6′/6”		168.4	>C=O
7/7′/7”	6.25, brs	117.4	-CH=
8/8′/8”		151.3	>C=
9/9′/9”	2.8, m	43.3	-CH_2_-
10/10′/10”	4.08, t (6.28)	64.8	-CH_2_-
11/11′/11”	1.86, s	21.9	-CH_3_

## Data Availability

Not applicable.

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
