# Peer review of "Genetic Engineering of Talaromyces marneffei to Enhance Siderophore Production and Preliminary Testing for Medical Application Potential"

_jof, 2022, doi:10.3390/jof8111183_

Round 1

Reviewer 1 Report

This work intended to enhance the siderophore production in Talaromyces marneffei by delete sreA. This idea is acceptable, although the design shows weak novelty. However, some issues should be addressed before it can be accepted for publication.

1.    The partially purified siderophore was analyzed by reverse-phase HPLC using absorbance determination at 434 (A) and 254 nm (B), whereas the figure legend in Figure 3A was indicated as Abs.435. Which one is right? What is Abs. 254 for? It was not defined. Moreover, according to the previous study (Lu et al., ACS Synthetic Biology, 2019, 8, 1755−1765; Lu et al., Functional & Integrative Genomics, 2019, 19, 137–150), the Abs.435 is for the HPLC detection of iron containing siderophores. In this work, whether the siderophores were saturated by Fe3+ was not specified. Please revise this point. As for the MS, the authors performed it in an ESI+ mode. This mode would always cast positive ions, e.g., H+, K+, Na+, on the fragments of N/O containing molecules, like Coprogen B in this work. However, this was not seen in the paper. Please explain this point.

2.    Figure 7. Coprogen B Structure Investigated by 1H-NMR and 13C-NMR spectroscopy. There is absence of explanation of Figure 7A\7B\7C.

3.    The Figure 9 is not in high quality, please replace it to a high-definition picture and scale bars should be added.

4.    Interestingly, the chelation efficiency of coprogen B was higher …….(%FI of coprogen B vs DFO at 50, 25, and 12.5 µg were 29.41 vs 24.41, 27.75 vs 22.25, 25.08 vs 20.25, respectively). This description is inconsistent with Figure 10C data, please check your manuscript. And the title of Figure 10A\10B is described unclearly. Please improve these points.

5.    In Introduction and Discussion sections, the genetic manipulation towards Aureobasidium melanogenum for enhancing siderophores, including deletion of sre1 (sreA) gene in Aureobasidium, should be cited to fully include the present proceedings of fungal siderophores. Please improve these two sections.

Reviewer 2 Report

The manuscript which describes the genetic modification of a Talaromyces marneffei fungus, the isolation and characterization of a siderophore the is well-written and could be (with minor edits) be accepted.

Some comments:

Line 124: more detail should be given on the PCR and especially RT-PCR methods that were used to confirm the deletion

Line 130: "inoculation" is better ti use than "cultivation"

Line 236: Which strains were used. It is not good just to say the species name

Line 241: What media was used to cultivate C. albicans?

Figure 1. Please give a scale (in the form of a bar) to indicate the sizes of the germinated spore. 12 days is quite long to see such little amount of growth?

Line 527: I think ALL organisms require iron and not just "most"

Line 537: Please avoid contractions like "it's"

Line 546-547: A preposition "of" is missing (i think).

Line 574: Can you give the catalytic function of sidG?

Round 2

Reviewer 1 Report

The revised manuscript can be accepted after examination of minor errors, e.g., typos, punctuation, capitalization.
